# Dynamic inflow model for a Floating Horizontal Axis Wind Turbine in surge motion

Carlos Ferreira[1], Wei Yu[1], Arianna Sala[1], and Axelle Viré[1]

[1]Delft University of Technology, Faculty of Aerospace Engineering, Kluyerweg 1, 2629 HS Delft, the Netherlands

**Correspondence:** Carlos Ferreira (c.j.simaoferreira@tudelft.nl)

**Abstract.** Floating Offshore Wind Turbines may experience large surge motions, which can cause blade-vortex interaction if they are similar to or faster than the local wind speed. Previous research hypothesized that this blade-vortex interaction phenomenon represented a turbulent wake state or even a vortex ring state, rendering the Actuator Disc Momentum Theory and the Blade Element Momentum Theory invalid. This hypothesis is challenged, and we show that the Actuator Disc Momentum Theory is valid and accurate in predicting the induction at the actuator in surge, even for large and fast motions. To accomplish this, we develop a dynamic inflow model that simulates the vorticity-velocity system and the effect of motion. The model's predictions are compared to other authors' results, a semi-free wake vortex-ring model, other dynamic inflow models, and CFD simulations of an actuator disc in surge. The results show that surge motion and rotor-wake interaction do not result in a turbulent wake or vortex ring state, and that the application of Actuator Disc Momentum Theory and Blade Element Momentum Theory is valid and accurate when applied correctly in an inertial reference frame. In all cases, the results show excellent agreement with the higher fidelity simulations. The proposed dynamic inflow model includes a modified Glauert's correction for highly loaded streamtubes and is accurate and simple enough to be easily implemented in most Blade Element Momentum models.

## 1 Introduction

### 1.1 Motivation for the research

Floating offshore wind turbines (FOWTs) are supported by floating foundations, resulting in greater motion than wind turbines supported by bottom-mounted foundations. (de Vaal et al., 2014). This increased freedom of motion can result in several unsteady aerodynamic phenomena at the airfoil, blade, rotor, and wake scales, as studied by Sebastian and Lackner (2012), Sebastian and Lackner (2013), Sivalingam et al. (2018), Kyle et al. (2020), Wen et al. (2017), Lee and Lee (2019), de Vaal et al. (2014), Mancini et al. (2020), Micallef and Sant (2015), Tran and Kim (2016), Chen et al. (2021), Shen et al. (2018), Lee and Lee (2019), Farrugia et al. (2016), Cormier et al. (2018), Dong et al. (2019), Dong and Viré (2021) and others.

The complexity of the aerodynamics resulted in many interpretations of the phenomena. Several authors proposed that the flow could change from windmill to propeller state due to motion and changes in loading. Furthermore, several authors proposed that if the surge velocity is large enough, the combination of wind speed and surge velocity would be less than twice

the induction velocity, resulting in a turbulent wake state or even a vortex ring state (see Sørensen et al. (1998) for the definition of turbulent wake state and vortex ring state). Actuator Disc Momentum Theory, according to many authors, would no longer be valid under these conditions. Due to the fact that Blade Element Momentum Theory (BEM, see Glauert (1935)) is based on Actuator Disc Momentum Theory, the occurrence of turbulent wake state and vortex ring state would significantly limit the use of BEM for FOWTs. Given that BEM is the most commonly used tool for simulating the aerodynamics of horizontal axis wind turbines (Madsen et al., 2020), this could have a significant impact on our design methods.

However, the prediction of turbulent wake state and vortex ring state for the actuator disc (wind turbine) in periodic surge motion appears to be in most cases the result of an invalid interpretation of the Actuator Disc Theory. As stated by Sørensen and Myken (1992), since the concept of the actuator disc was first formulated by Froude it has been closely related to the one-dimensional momentum theory and much confusion about its applicability in describing complex flow fields still exists. This is particularly true for the case of an actuator in cyclic motion, as is the case of FOWTs.

The name of the theory is in itself misleading, because the Actuator Disc Momentum theory is in fact the theory of the mass and momentum balance of the streamtube that includes the actuator. The actuator disc is a physical model that enables a discontinuity of the pressure field into the governing flow equations as the reaction to an external force field. The added information that the pressure discontinuity occurs at the actuator allows us to estimate the velocity at the actuator by evaluating stagnation pressure along the streamtube. Therefore, Actuator Disc Theory refers to the state of the streamtube defined in an inertial reference frame that contains the actuator which is static in the same inertial reference frame. Propeller state, windmill state, turbulent wake state, vortex ring state and propeller brake state do not refer to the state of the actuator but to the state of the streamtube (Sørensen et al., 1998). In an unsteady flow, an actuator might have an instantaneous loading as a propeller, while the streamtube remains in windmill state. Two examples of such inertial reference frames are the one attached to the steady streamtube which includes the actuator disc associated with a stationary wind turbine (or propeller) in an incoming unperturbed wind speed $U_\infty$ of any value, or the one attached to the steady streamtube that contains an actuator disc in a constant motion (not accelerated) in an incoming unperturbed wind speed $U_\infty$ of any value.

When the actuator is moving in an inertial reference frame with a steady velocity, the streamtube and actuator are in the same inertial reference frame, and the reference unperturbed velocity of the wind used in the actuator disc model $U_{\infty_{ref}}$ is the sum of the velocity of the wind in the inertial reference frame $U_\infty$ with the moving velocity of the actuator in the inertial reference frame $v_{act}$, as

$$U_{\infty_{ref}} = U_\infty - v_{act} \tag{1}$$

In the condition that $v_{act}$ is constant (time invariant), the actuator disc momentum theory applies, and the thrust coefficient $C_T$ is defined as

$$C_T = \frac{T}{\frac{1}{2}\rho U_{\infty_{ref}}^2 A} = 4a\left(1-a\right) \tag{2}$$

where $T$ is the thrust applied by the actuator, $A$ is the area of the actuator and $a$ is defined as the induction factor, such that the velocity perceived by the actuator $U_{act}$ (at the location of the actuator) is given by

$$U_{act} = (1-a)\,U_{\infty_{ref}} \tag{3}$$

Strictly speaking, the Actuator Disc Theory cannot be applied to a non-inertial reference frame (e.g. the actuator disc in an arbitrary or periodic surge motion) as this violates the steady assumption. The transition to the accelerated reference frame of the actuator requires the addition of apparent forces in the momentum equation, which are not accounted in the Actuator Disc Momentum theory. Therefore, for FOWTs experiencing accelerated motions, Equation 1 to 3 are invalid for predicting the induction at the oscillating actuator using 1D momentum theory.

Another common misconception is that a perceived negative velocity at the actuator (e.g. the actuator moving downwind faster than the wind during the oscillatory surge motion) represents a vortex ring state. However, the vortex ring state is a property of the streamtube, evaluated in the inertial reference frame of the streamtube. If there is no flow reversal in the streamtube, there is no vortex ring state. For an interpretation of vortex ring state see the works of Sørensen et al. (1998) and Sørensen and Myken (1992). Equally, although the load on the actuator can range from negative (propeller) to highly loaded, that does not mean that the streamtube will vary from propeller state to turbulent wake state. If the oscillation of the loading is very fast, the flow does not have enough time to accelerate and the streamtube will remain in windmill state.

Although the actuator disc model is one-dimensional and assumes steady, incompressible and inviscid flow, when used in engineering applications in unsteady flow, the steady assumption is relaxed and the model can be corrected by dynamic inflow models. If a dynamic inflow model could solve the streamtube induction and the induction at the location of the actuator, BEM could then be used for the simulation of FOWTs. The motivation of this work is to achieve this goal.

## 1.2 Aim of the research and rationale for model derivation

The aim of the research is to:

1. Derive and apply a dynamic inflow model as a correction for the effect of surge on the estimation of the induction at the actuator disc.

2. Validate the approach by comparison with the results of higher fidelity models, namely potential flow vortex ring simulations and CFD simulations.

3. Demonstrate that, for the cases investigated here (including cases with large surge velocities and loading), turbulent wake state and vortex-ring state do not occur as a consequence of the surge motion, and therefore BEM is still valid.

The model is derived using the following rationale. The surging actuator disc generates an unsteady flow, which violates the actuator disc model's assumption of steady flow. It is difficult to solve the unsteady momentum equation in an inertial or non-inertial reference frame using pressure-velocity solutions. However, whether the reference frame is accelerated or inertial, a

lagrangian formulation of wake generation and convection and the resulting vorticity-velocity system solution of the induction field are invariant. A dynamic inflow model inspired by the lagrangian vorticity distribution should accurately predict the induction at the actuator, as demonstrated by Yu et al. (2019a) and Yu (2018). The wake and induction solutions are linear superpositions of a newly released wake (new wake) and a previously released wake (old wake), with respect to the reduced time scale of the flow. The dynamic inflow models by Øye (1986), Larsen and Madsen (2013), Yu (2018) and Madsen et al. (2020) implicitly model this superposition and convection of the vorticity system, while explicitly defining the wake length and wake convection speed across time scales; these models should serve as a foundation for developing the proposed model. The actuator's displacement dynamics can be interpreted as changing the vorticity system's relative convection speed, as it is invariant with respect to the reference frame. The quasi-steady solution for a fully developed wake with the strength of newly shed wake elements can be determined using a modified 1D steady actuator disc model that simulates wake generation and convection caused by the force field. This 1D actuator disc model with dynamic inflow should be comparable to solutions from higher fidelity models, such as prescribed and (semi-) free-wake vortex-ring models, or CFD simulations.

In Section 1.3 we define the surge motion and thrust functions. Section 1.4 presents a summary of study cases found in literature, organised in distributions of the range of parameters that define the surge motion and thrust function.

## 1.3  Description of the motion of the actuator and loading on the actuator

The simulations and analysis in this work use the following assumptions. The actuator surface is a circle of diameter $D$ (radius $R = D/2$), and is always normal to the unperturbed free-stream $U_\infty$. The latter is uniform, steady, and aligned with the $x$-direction. The actuator moves in the $x$-direction according to Equation 4, where $x_{act}$ is the location of the actuator in the $x$-axis, $A_{x_{act}}$ is the amplitude of the motion and $\omega$ is the frequency of the motion, defined in relation to a reduced frequency $k$ as stated in Equation 5. The loading over the actuator is uniform and normal to the surface, and the thrust coefficient $C_T$ is defined by Equation 6 taking $U_\infty$ as reference for the dynamic pressure, where $C_{T_0}$ is the average thrust coefficient, $\Delta C_T$ is the amplitude of the variation of $C_T$, $\phi$ is an additional phase difference between motion and loading, and $t$ represents time. The sinusoidal loading approximates the load oscillations observed by other authors, as described in Section 1.4. The load change is a first-order result of the sinusoidal change in the non-entry boundary condition on the blades/actuator surface caused by the sinusoidal motion (this is further expanded in Section 1.4).

$$x_{act} = A_{x_{act}} \sin\left(\frac{kU_\infty}{D} t\right) \tag{4}$$

$$k = \frac{\omega D}{U_\infty} \tag{5}$$

$$C_T = \frac{T}{\frac{1}{2}\rho U_\infty^2 A} = C_{T_0} - \Delta C_T \cos\left(\frac{kU_\infty}{D} t + \phi\right) \tag{6}$$

## 1.4 Survey of study cases in previous experimental and numerical research

Figure 1 presents a survey of the experimental and numerical study cases in the work of de Vaal et al. (2014), Kyle et al. (2020), Mancini et al. (2020), Micallef and Sant (2015), Tran and Kim (2016), Chen et al. (2021), Sivalingam et al. (2018), Shen et al. (2018), Lee and Lee (2019), Farrugia et al. (2016), Wen et al. (2017), Cormier et al. (2018) and Dong et al. (2019). The results are organised in $\frac{A_{x_{act}}}{D}$ vs. $k$ with isocurves of $v_{max}$ in Figure 1a, $\Delta C_T$ vs $v_{max} = \frac{\omega A_{x_{act}}}{U_\infty}$ in Figure 1b and $\Delta C_T$ vs. $C_{T_0}$ in Figure 1c . Orange symbols represent Eulerian Navier-Stokes simulations (commonly referred to as CFD), green symbols represent Lagrangian vortex models, and blue symbols represent experiments (some also including simulations). Figure 1b is inspired by the work of Mancini et al. (2020). The survey shows that amplitudes of the motion are below $0.13D$ and reduced frequency $k < 15$. More importantly, the maximum surge velocity is $v_{max} < 1.15$. The relation of $\Delta C_T$ to $C_{T_0}$ shows that only in three cases the thrust reaches negative values. The almost linear relation of $\Delta C_T$ to $v_{max}$ confirms the earlier observations by Mancini et al. (2020). An hypothesis is that the linear relation is explainable by the linear effect between the surge velocity and the circulation on the blades, due to the change of the non-entry boundary condition on the blade surface. This hypothesis is expressed by the Equation 7, in which we consider the two-dimensional thrust coefficient at a given blade section. $a'$ azimuthal induction is omitted. The aerodynamics of the blade section are approximated using a potential flow flat plate formulation. The change in section thrust $\Delta C_{T_{blade\ section}}$ is then a function of the change in circulation $\Delta \Gamma$ and the rotor's local azimuthal velocity $\lambda_r U_\infty$ at radial position $r$ (we disregard added mass effects). The change in circulation is a function of the chord $c$ of the section and the non-entry boundary condition, which is defined as the internal product of the section's normal $\overrightarrow{n}$ and the change in axial velocity $\Delta \overrightarrow{v}_{axial}$, the thrust variation equation is expressed as a function of the local variation of axial velocity, which is dominated by the surge motion.

$$\Delta C_{T_{blade\ section}} = \frac{\lambda_r \Delta \Gamma}{r \pi U_\infty} = \frac{\lambda_r \overrightarrow{n} \cdot (\Delta \overrightarrow{v}_{axial})\ c\pi}{r \pi U_\infty} \approx \frac{c}{r} \lambda_r \frac{\Delta v_{axial}}{U_\infty} \tag{7}$$

In this work we will evaluate the proposed Actuator Disc Momentum theory with dynamic inflow correction in a motion and load space wider than (and encompassing) the one in Figure 1. The next section presents the Methods used in the research. It is followed by the Results and Discussion and finally the Conclusions.

## 2 Methods and approach

The results presented and discussed in the Section Results and Discussion have five sources: the Navier-Stokes simulations of an actuator disc in surge by de Vaal et al. (2014); simulations by a semi-free wake vortex-ring model of an actuator disc in surge motion developed in this work; dynamic inflow models derived by other authors; CFD simulations of an actuator disc with imposed thrust; and a 1D Actuator Disc Momentum model corrected for the unsteady surge motion and loading by using a dynamic inflow model derived in this work. The cases are defined by the surge motion and unsteady load on the actuator. The results and discussion compare the estimated induction at the actuator disc. The higher fidelity results (Sections 3.1, 3.2

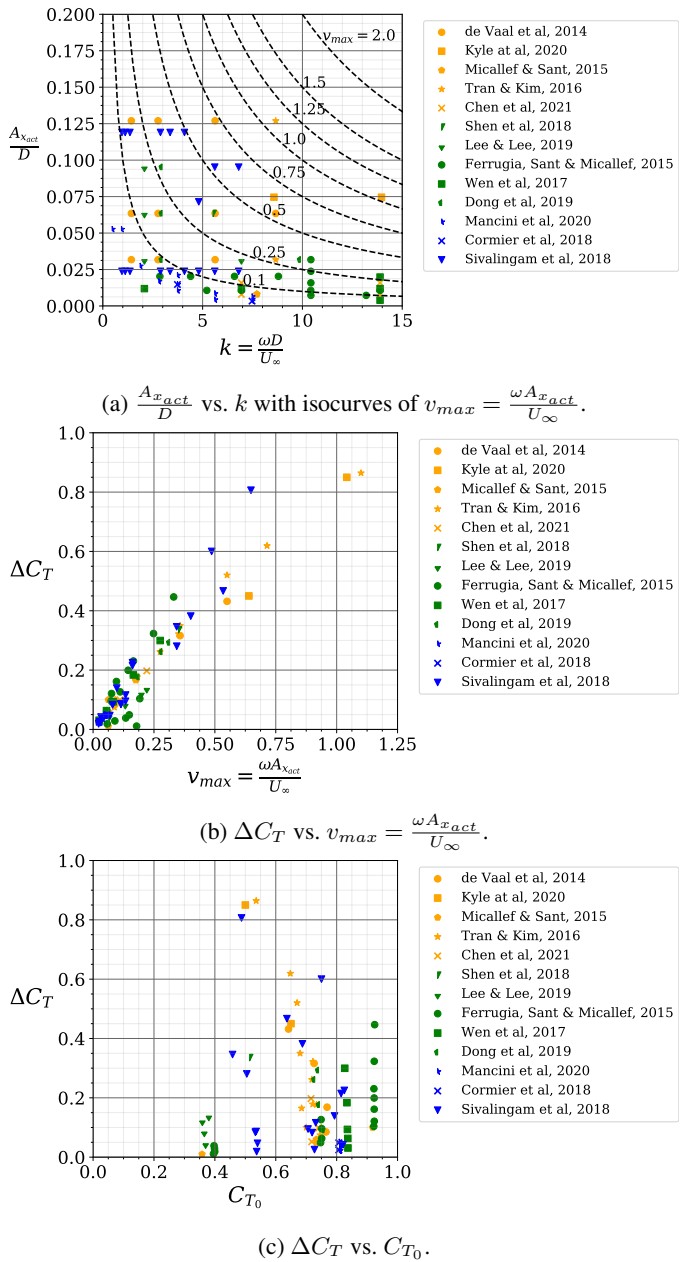

(a) $\frac{A_{x_{act}}}{D}$ vs. $k$ with isocurves of $v_{max} = \frac{\omega A_{x_{act}}}{U_\infty}$.

(b) $\Delta C_T$ vs. $v_{max} = \frac{\omega A_{x_{act}}}{U_\infty}$.

(c) $\Delta C_T$ vs. $C_{T_0}$.

**Figure 1.** Survey of the experimental and numerical study cases in the work of de Vaal et al. (2014), Kyle et al. (2020), Mancini et al. (2020), Micallef and Sant (2015), Tran and Kim (2016), Chen et al. (2021), Sivalingam et al. (2018), Shen et al. (2018), Lee and Lee (2019), Farrugia et al. (2016), Wen et al. (2017), Cormier et al. (2018) and Dong et al. (2019). The study cases are organised according to the key operational indicators: $\frac{A_{x_{act}}}{D}$ vs. $k$ with isocurves of $v_{max}$ (sub-figure $a$)), $\Delta C_T$ vs $v_{max} = \frac{\omega A_{x_{act}}}{U_\infty}$ (sub-figure $b$) and $\Delta C_T$ vs. $C_{T_0}$ (sub-figure $c$) ). Orange symbols represent Eulerian Navier-Stokes simulations (commonly referred to as CFD), green symbols represent Lagrangian vortex models, and blue symbols represent experiments (some also including simulations).

and 3.4) are used as benchmark for the results of the proposed dynamic inflow model. The impact of actuator motion is also demonstrated by comparing the proposed dynamic inflow model to other dynamic inflow models (Section 3.3).

## 2.1 Semi-free wake vortex ring model

The semi-free wake vortex ring model is a conventional model inspired by the approaches in the works of Yu et al. (2016), Yu (2018), van Kuik (2018) and van Kuik (2020). The "semi-free wake" description is due to the fact that the wake expands and convects with self induction up to five diameters downstream of the actuator. After that location, the expansion is frozen and the wake convects with a velocity based on $U_\infty$ and the velocity at the center of the wake.

## 2.2 CFD actuator disc model

*OpenFOAM* (OpenFOAM) was used to create the CFD actuator disc model. To reduce computational cost, a 3D computational domain with the shape of a parallelepiped is created and the hypothesis of axisymmetric flow is used. The velocity and pressure boundary conditions are imposed at the inlet and the outlet, respectively. The symmetry boundary conditions are imposed on one side and the bottom of the domain, and slip-wall boundary conditions are imposed on the other side and the top of the domain. A domain independency study is used to determine the dimensions of the domain. The mesh is dense around the actuator disc and becomes coarser as it moves away from it. A mesh independency study is used to determine the size of the cells surrounding the actuator disc. With a turbulence intensity of $0.1\%$, the RANS $k - epsilon$ turbulence model is used. $5e+06$ is the Reynolds number. It is demonstrated that the chosen turbulence intensity and Reynolds number have no significant effect on the outcome (Sala, to be published in December 2021).

The loading is applied using the Equation 6 and is uniformly distributed over the actuator disc, whose position varies over time using the Equation 4. The disc average axial induction factors obtained with steady CFD simulations are compared to those predicted by momentum theory with Glauert correction for thrust coefficients ranging from $CT = 0.2$ to $CT = 1.2$ to validate the model. The results are depicted in Figure 2. The results agree well with momentum theory at low thrust coefficients. At low thrust coefficients, the results agree well with momentum theory. The difference is $2.1\%$ at $C_T = 0.8$, and it grows larger as the thrust coefficient increases.

## 2.3 Dynamic inflow models by other authors

In this paper, we compare the results of induction using the proposed dynamic inflow model and five previously published dynamic inflow models. The five models are Pitt and Peters (1981) as described by Yu (2018), by Øye (1986) as described by Yu (2018), the model by Larsen and Madsen (2013), Yu (2018) (also described by Yu et al. (2019b)) and Madsen et al. (2020). The results of the models are labeled *Pitt-Peters*, *Øye*, *Larsen-Madsen*, *Yu* and *Madsen* in the figures of Section 3.3. The new dynamic inflow model presented in this work is labelled as *Ferreira*. The reader is also directed to the *ECN model* (see Schepers (2012)), which expands on the model developed by Pitt and Peters (1981).

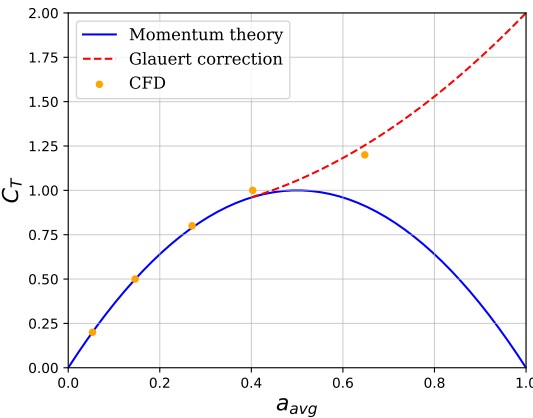

**Figure 2.** Disc average induction factor $a_{avg}$ against $C_T$ calculated with CFD actuator disc model and momentum theory with Glauert correction.

## 2.4  Formulation of the new dynamic inflow model including actuator motion

Section 1 presented the rationale for the formulation of the new dynamic inflow model including actuator motion. The aim is to simulate the dynamics of the vorticity-velocity solution of induction at the actuator. The approach of a convolution of quasi-steady solutions was proven effective by Øye (1986), Larsen and Madsen (2013), Yu et al. (2019a) and Yu (2018), and Madsen et al. (2020). This was often approached as a convolution of quasi-steady solutions of the 1D actuator disc theory or unsteady langragian solutions of step changes in the momentum balance. These models were then calibrated to the time and

length scales of the impulse responses (e.g. Yu (2018)). From the different formulations, the one of superposition of exponential decay of solutions (as e.g. presented by Larsen and Madsen (2013)) lends itself best to our objective of an explicit description of the invariant solution. For reference, the work of Madsen et al. (2020) presents an updated version of the Madsen/Larsen-Madsen dynamic inflow model (Larsen and Madsen (2013)), following up on the work by Pirrung and Madsen (2018). The Madsen dynamic inflow model is conceptualized as a curve fit of the solution of an unsteady actuator disc in a step function that

uses two time scales to better approximate the radial dependency of the unsteady induction, implicitly as a near wake and far wake time scales. This is also the interpretation proposed in the work of De Tavernier and Ferreira (2020) when reviewing the implementation for Vertical Axis Wind Turbines (see also Larsen and Madsen (2013)), discussing the time scales as near wake and far wake. The model presented by Pirrung and Madsen (2018) predicts several corrections for loading and radial effects and is calibrated against higher fidelity simulations. The two time constant filter approach was previously proposed by Øye

(1986), and represents a departure from the approach by Pitt and Peters (1981) of the solution of the pressure-velocity towards the solution of vorticity-velocity problem. This solution of the vorticity-velocity problem was discussed by Øye (1986), Larsen and Madsen (2013) and Madsen et al. (2020) as a dynamic filter of near and far wake solutions.

In this work we take inspiration of the two time scales approach for representing the contribution of the wake generated previously and the newly shed wake, and to distinguish between the induction at streamtube scale from the induction at the

actuator. The solution of the vorticity-velocity system does not require the time integration of the flow acceleration, but it is calculated directly from the vorticity system at each time step. The wake solution and the induction solution are the linear superposition of a newly released wake (new wake) and a previously released wake (old wake), in relation to the reduced time scale of the flow. The convection of the two wake systems must be determined. We therefore define two reference values of induction, namely the streamtube induction velocity $u_{str}$ and the induction velocity at the location of the actuator $u_{act}$. We use these velocities to determine the convection of the vorticity system in the streamtube and in relation to the actuator.

The first variable of the dynamic inflow model is the unperturbed reference velocity on the inertial reference frame that contains the streamtube and the actuator. In the case of the actuator in an oscillating surge, the reference velocity can be defined as in Equation 8

$$U_{\infty_{ref}} = U_{\infty} \tag{8}$$

The second variable of the model is the streamtube wake-convection reference velocity, as defined in Equation 9. $U_{str}$ is determined by averaging the two induction terms $u_{str}$ and $u_{act}$; the equal weighing of the two induction terms reflects the balance between the proximity of the short newly shed wake to the region where the velocity is evaluated (actuator) and the distance to the longer previously shed vorticity system. Although different averaging weights can lead to more fine tuned solutions, this relation appears to be sufficiently accurate.

$$U_{str} = U_{\infty_{ref}} - \frac{u_{str} + u_{act}}{2} \tag{9}$$

We can calculate an equivalent quasi-steady solution of the induction velocity of a vorticity system generated by a thrust $C_T$ and wake convected in streamtube with reference velocity $U_{str}$ (Equation 10) to be later used as a forcing function of a steady solution of the newly shed wake. It is important to note that this forcing function differs from the one commonly used in dynamic inflow models (usually the steady induction for a given thrust coefficient as defined in Equation 2). Equation 10 approaches the 1D steady actuator disc thrust equation, taking $U_{str}$ as the mass flow rate that experiences a momentum change of $u_{qs}$ (per unit fluid density). If the system converges to a steady flow, Equation 10 converges to Equation 2.

$$u_{qs} = \frac{C_T U_{\infty}^2}{4} \frac{1}{U_{str}} \tag{10}$$

We can choose to apply a form of Glauert's correction for the case of heavily loaded streamtubes and instantaneous $C_T > 0$, inspired in the formulation presented by Burton et al. (2011). The heavily loaded streamtube criterion is defined as

$$U_{str} > U_{\infty_{ref}} \left(1 - \frac{\sqrt{C_{T_1}}}{2}\right) \tag{11}$$

with $C_{T_1} = 1.816$.

If the criterion in Equation 11 applies, the value of $u_{qs}$ can be determined by Equation 12, curve fitted from Glauert's correction as described by Burton et al. (2011).

$$u_{qs} = -1.883 - 1.540\sqrt{\frac{C_T U_\infty^2}{4}\frac{1}{U_{str}}} + 4.086\sqrt[4]{\frac{C_T U_\infty^2}{4}\frac{1}{U_{str}}} \tag{12}$$

Due to the fact that wake convection varies along the streamtube, we now define length scales for actuator/near wake $L_{act}$ and streamtube/far wake scale $L_{str}$ in Equations 13 and 14. The choice of one and five diameters are suitable for near and far wake scales ; at one diameter the wake has achieved over 90% of its expansion and increase in induction, and the vorticity in the first five diameters accounts for over 99% of the solution of induction at the actuator. The choice for integer values of length scales is somewhat arbitrary; in the development of other dynamic inflow models, authors have fine tuned these scales

to improve matching with the solution of impulse flow. In this model, slightly changing these scales to other similar values will not significantly affect the results of the model. The length scales are defined as half of the near and far wake scales for application in the exponential functions of the time integration and filter functions.

$$L_{act} = \frac{1}{2}1D \tag{13}$$

$$L_{str} = \frac{1}{2}5D \tag{14}$$

We now define time scales of convection of the wake for actuator/near wake scale and streamtube/far wake scale. For the streamtube scale we define one time scale $\tau_{str}$ given by Equation 15, used for the convection of the old vorticity system and the convection of the generation of the new vorticity system.

$$\tau_{str} = \frac{L_{str}}{U_{\infty_{ref}} - \frac{u_{str}}{2}} \tag{15}$$

     For the actuator/near wake scale we need to define two time scales: one for the convection of the old vorticity system

(Equation 16) and another for the convection of the generation of the new vorticity system (Equation 17). The velocity of the actuator is defined as the time derivative of position of the actuator $v_{act} = \dfrac{\mathrm{d}x_{act}}{\mathrm{d}t}$

$$\tau_{act_1} = \frac{L_{act}}{U_\infty - \frac{u_{act}}{2} - v_{act}} \tag{16}$$

$$\tau_{act_2} = \frac{L_{act}}{U_{\infty_{ref}} - \frac{u_{act}}{2}} \tag{17}$$

     Following the approach by Larsen and Madsen (2013), we can now calculate the new solutions of the streamtube induction

velocity $u_{str}$ and the induction velocity at the location of the actuator $u_{act}$ by the implicit integration in time of the effect of

the filtered forcing function $u_{qs}$. The approach is similar to that of Øye (1986) which, however, has an explicit integration in time of the filtered forcing function.

$$u_{act_{(t+\Delta t)}} = u_{act_{(t)}} e^{-\frac{\Delta t}{\tau_{act_1}}} + u_{qs}\left(1 - e^{-\frac{\Delta t}{\tau_{act_2}}}\right) \tag{18}$$

$$u_{str_{(t+\Delta t)}} = u_{str_{(t)}} e^{-\frac{\Delta t}{\tau_{str}}} + u_{qs}\left(1 - e^{-\frac{\Delta t}{\tau_{str}}}\right) \tag{19}$$

When $U_{\infty_{ref}} = U_\infty$, Equation 18 can also be written as Equation 20.

$$u_{act_{(t+\Delta t)}} = u_{act_{(t)}} e^{-\frac{\Delta t}{\tau_{act_2}}} e^{\Delta t \frac{v_{act}}{L_{act}}} + u_{qs}\left(1 - e^{-\frac{\Delta t}{\tau_{act_2}}}\right) \tag{20}$$

Equation 20 shows the effect of the actuator motion ($v_{act}$ is defined in the same reference frame as $U_\infty$). As the actuator moves away from the previously shed wake, the effective induction decreases. As the actuator moves into the wake, the effective induction increases.

The model can be generalised to the case of actuator motions that have a non-zero average displacement, e.g. an actuator travelling in forward motion with periodic oscillations. In this case, the most suitable inertial reference frame needs to be updated and so does $U_{\infty_{ref}}$. The varying reference wind speed can be determined by Equation 21

$$U_{\infty_{ref(t+\Delta t)}} = U_{\infty_{ref(t)}} e^{-\Delta t \frac{U_{\infty_{ref(t)}}}{L_{str}}} + (U_\infty - v_{act})\left(1 - e^{-\Delta t \frac{U_{\infty_{ref(t)}}}{L_{str}}}\right) \tag{21}$$

An example of the implementation of the model as an algorithm in Python is shown in *Appendix A: Implementation of the*
*model as an algorithm in Python*.

In the Results section, the induction at the actuator is represented by its non-dimensioned form $a$, defined by Equation 22.

$$a = \frac{u_{act}}{U_{\infty_{ref}}} \tag{22}$$

## 3    Results and Discussion

### 3.1    Comparison of results of the dynamic inflow model with those of de Vaal et al. (2014).

This section compares the results of the dynamic inflow model with the results of the semi-free wake model and the results published in de Vaal et al. (2014), page 117, for a moving actuator disc modelled in the commercial software FLUENT using a finite volume discretization of the incompressible Navier-Stokes equations. The study case is an actuator disc in a sinusoidal surge motion and varying thrust. The four sub-cases have the same motion amplitude but four different motion frequencies.

Figure 3 describes the four sub-cases and presents the thrust curve and the resulting values of induction coefficient over the rotation. The location of the actuator $x_{act}$ is also plotted.

There are two important differences between the simulations in this work and the ones in de Vaal et al. (2014). The simulations with the dynamic inflow model and with the semi-free wake model use an unsteady uniform loading over the actuator, and the inductions plotted in Figure 3 correspond to the induction at the actuator at different radial positions. de Vaal et al. (2014) applied a rotor model (NREL 5MW) in their model, leading to an non-uniform loading. Additionally, the induction plotted in Figure 3 is the area weighted induction at the blade, including Prandtl's tip correction for finite blade effects. The non-uniform loading considered by de Vaal et al. (2014) and the inclusion of Prandtl's tip correction leads to a higher value of induction in relation to the average induction over the annulus. By studying the solution for the steady load case presented in the work of de Vaal et al. (2014) (Figure 4, page 112), it is possible to estimate the average induction using their approach to be between $a = 0.274$ and $a = 0.285$ (depending on tip correction model), while an actuator disc with uniform load and the same thrust coefficient ($C_T = 0.76$) will result in an induction of $a = 0.256$. This results in a $\Delta a \approx 0.023$ between the two methodologies. $a$ and $C_T$ are, as in the remaining of this work, defined in relation to the unperturbed wind speed $U_{\infty ref} = U_\infty$.

To support the interpretation of the results in Figure 3, Table 1 presents for each sub-case (labelled by the reduced frequency $k$) the average thrust coefficient $\overline{C_T}$, the amplitude of the variation of thrust coefficient $\Delta C_T$, the time average of the area-weighted and Prandtl-tip-corrected average induction $\overline{\overline{a}_{deVaal}}$, the time average of the area-weighted average induction obtained with the semi-free wake vortex ring model $\overline{\overline{a}_{sfwm}}$, the time average of the induction at the center of the actuator predicted by the dynamic inflow model $\overline{a}_{dynamic\ inflow}$, the time average of the induction calculated using steady Actuator Disc Theory $\overline{a_{(C_T)}}_{steady}$ and the steady induction of the time average of thrust coefficient $a_{(\overline{C_T})steady}$ (the last two predicted using steady 1D actuator disc theory).

The results in Figure 3 and Table 1 show that:

1. Comparing the results of de Vaal et al. (2014) and the vortex ring model, despite the difference of what is modelled (non-uniform loading vs. uniform loading) and the difference of the nature of the two values of induction (impact of Prandtl's tip correction), it results that $\overline{\overline{a}_{deVaal}} - \overline{\overline{a}_{sfwm}} < 0.02$.

2. Although the dynamic inflow model is one-dimensional, the difference to the semi-free wake vortex ring model prediction is, in all cases, less than $\Delta a < 0.01$ for the region $r/R \leqslant 0.8$.

3. With increasing reduced frequency, there is an increased phase shift between the curve of the motion/thrust and the resulting induction. The dynamic inflow model is able to capture the phase shift, matching what is observed in the results of de Vaal et al. (2014) and of the vortex ring model.

4. The results confirm that with increasing reduced frequency the average induction will differ from $\overline{a_{(C_T)}}_{steady}$ towards $a_{(\overline{C_T})steady}$ despite the higher amplitude $\Delta C_T$, a consequence of the inertia of the streamtube.

The results of the semi-free wake vortex-ring model show a larger oscillation of induction closer to the actuator edge. This is not a finite-blade tip effect, nor the radial variation of induction previously found in a steady actuator disc with uniform loading

**Table 1.** Table of averages of the results of Figure 3.

| $k$ | $\overline{C_T}$ | $\Delta C_T$ | $\overline{a}_{deVaal}$ | $\overline{a}_{sfwm}$ | $\overline{a}_{dynamic\ inflow}$ | $\overline{a_{(C_T)steady}}$ | $a_{\overline{(C_T)steady}}$ |
|------|------|------|------|------|------|------|------|
| 1.43 | .77 | .09 | .286 | .268 | .262 | .264 | .261 |
| 2.77 | .77 | .17 | .282 | .267 | .261 | .267 | .259 |
| 5.62 | .75 | .31 | .272 | .258 | .255 | .272 | .25 |
| 8.66 | .69 | .43 | .258 | .239 | .236 | .254 | .222 |

(van Kuik, 2018). It is actually an effect of blade (actuator) vortex interaction due to the motion of the actuator and unsteady loading.

The results listed above allow us to conclude that for this case study: 1) the semi-free wake vortex ring model provides results in excellent agreement with those of the higher fidelity model used by de Vaal et al. (2014); 2) the predictions of the dynamic inflow model are in excellent agreement with the results of the semi-free wake vortex ring model; 3) accounting for the $\Delta a$ due to the differences between non-uniform loading vs. uniform loading, the predictions of the dynamic inflow model are in excellent agreement with the results by de Vaal et al. (2014).

In the next section we will compare the predictions of the dynamic inflow model with the results of the semi-free wake vortex ring model for a more diverse and more challenging set of cases.

### 3.2 Comparison of results of the dynamic inflow model with those of the semi-free wake vortex ring model

In this section we present and discuss the comparison of the results of induction by the semi-free wake vortex ring model and the proposed dynamic inflow model for a sinusoidal surge motion with $x_{act} = A_{x_{act}} \sin(kU_\infty/Dt)$ (also plotted) and with $C_T = C_{T_0} - \Delta C_T \cos(kU_\infty/Dt)$, where the loading is uniformly distributed over the actuators.

Figure 4 presents the cases for $x_{act} = 0.1D \sin(kU_\infty/Dt)$ and $C_T = 0.5 - 0.5 \cos(kU_\infty/Dt)$ for six values of reduced frequency $k = [1.0, 3.0, 5.0, 10.0, 15.0, 20.0]$. The results show an excellent agreement between the semi-free wake vortex ring model and the proposed dynamic inflow model. The agreement improves with increasing reduced frequency. The model is also able to capture the progressive phase shift of the induction with increased reduced frequency, as the effect of the motion starts to dominate over the effect of varying thrust. Despite the large amplitude of loading and motion, the highest difference occurs in the case of lowest frequency, with the difference at some points of the cycle being $\Delta a = 0.02$. In this low frequency, the streamtube is significantly accelerated due to the slowly changing load, and the dynamic inflow model must capture this acceleration.

Figure 5 allows us to distinguish the effect of motion from the effect of varying thrust. Figures 5a and 5b allow to compare the effect of increasing the reduced frequency of the motion while the thrust remains constant. Due to motion, the induction is higher when the actuator is in the downwind region (the actuator moves faster than the wake and immerses in its own wake), and lowers as the actuator moves upwind (lower density of vorticity in the near wake). The increasing frequency of motion increases the amplitude of the induction and shifts its phase. Although it shifts towards the phase of the position of the motion,

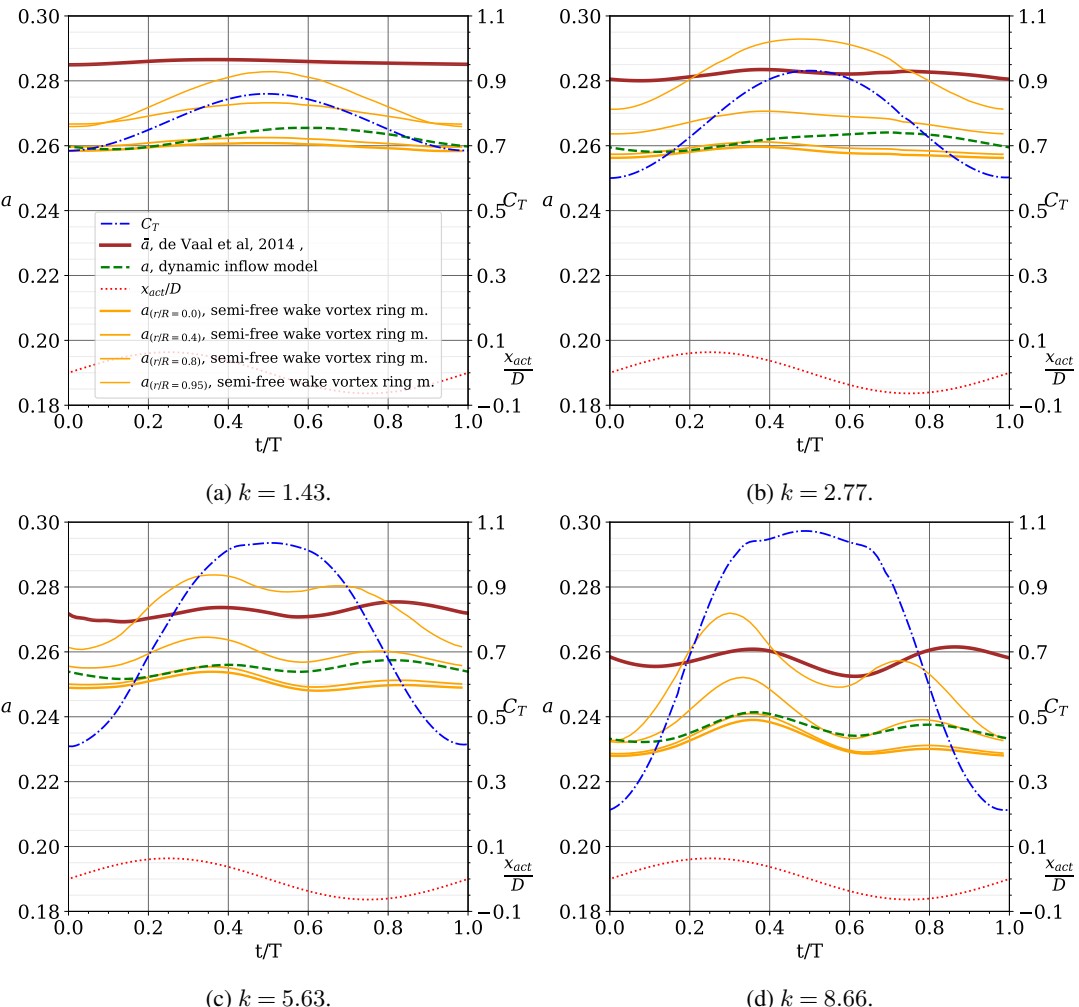

**Figure 3.** Comparison of the results of induction by de Vaal et al. (2014) ($\bar{a}$ average induction factor over the actuator), the semi-free wake vortex ring model ($a$ at different radial positions $r/R$) and the new proposed dynamic inflow model. The four case studies are defined by a surge motion of the actuator a sinusoidal motion with $x_{act} = A_{x_{act}} \sin(kU_\infty/Dt)$ with $A_{x_{act}} = 0.063D$, and $k = 1.43, 2.77, 5.63$ and 8.66. The resulting thrust coefficient $C_T$ is also plotted. The results are plotted over one period, along the non-dimensioned time $t/T$. All values are non-dimensioned with relation to $U_\infty$.

it is actually shifting towards a $\pi/2$ shift in relation to the velocity of the motion. Figures 5c and 5d show the cases of a static actuator where the load is phase shifted by $\pi$ between the two figures. The inductions are naturally also phase shifted by $\pi$. Although trivial, these two cases are important to understand Figures 5e and 5f. Figure 5e corresponds to the typical case experienced by a surging wind turbine, where the loading is highest when the actuator moves upwind and lowest when the actuator moves downwind. The effects of motion on the near wake density and the effects of thrust are out of phase and mostly cancel each other. Figure 5f shows a case that is mostly infeasible in a floating wind turbine (and probably undesirable as it could be unstable), where the thrust and motion are in phase and accumulate. This theoretical case allows us to push the dynamic flow model to one of the more challenging cases as it results in a larger amplitude of induction. However, even in this case, the dynamic inflow model is in good agreement with the results of the semi free wake model.

Figure 6 shows the comparison of the two models for six cases where the amplitude of thrust is proportional to the maximum surge velocity $\Delta C_T = \frac{kA_{x_{act}}}{D}$. The values of amplitude of the motion is the same for all cases $A_{x_{act}} = 0.1D$. The six value of reduced frequency are $k = [1.0, 3.0, 5.0, 10.0, 15.0, 20.0]$ implying $\Delta C_T = [0.1, 0.3, 0.5, 1.0, 1.5, 2.0]$, while the average thrust coefficient is $C_{T_0} = 0.8$. The results show that the increased speed of motion mostly cancels the effect of the varying thrust, and the induction remains almost constant. The two models are in excellent agreement in the prediction of the induction (the difference is below $0.02$ in all cases). The increased frequency leads to higher changes of loading, but the variation is so fast that the streamtube does not change the velocity significantly.

### 3.3 Comparison of results of the dynamic inflow model with those of other dynamic inflow models

In this section, we compare the results of induction using the semi-free wake vortex ring model to those of the proposed dynamic inflow model and five previously published dynamic inflow models. In the results of Figure 7, the induction is evaluated in the inertial reference frame. For the *Pitt-Peters*, *Øye*, *Larsen-Madsen*, *Yu* and *Madsen* models, the motion of the actuator cannot be taken into account. Only the *Ferreira* model accounts for the motion of the actuator. The cases in Figure 7 cover several combinations of motion and thrust. The *Pitt-Peters*, *Øye*, *Larsen-Madsen*, and *Madsen* models were modified to account for Glauert's correction for heavily loaded actuator in their quasi-steady forcing function term.

The findings corroborate previous discussions. For non-moving actuators (Figures 7a and 7c), the various dynamic flow models agree reasonably well, with the more advanced/complex models (*Yu*, *Madsen*, and *Ferreira*) agreeing better with the semi-free wake vortex ring model results. The agreement between models decreases as the average $C_T$ and reduced frequency $k$ increase between Figures 7a and 7c. Due to the fact that the *Pitt-Peters*, *Øye*, *Larsen-Madsen*, *Yu* and *Madsen* models do not account for actuator motion, their results differ from those of the semi-free wake vortex ring model for the cases shown in Figures 7b, 7d, 7e, and 7f. Because the *Madsen* time scale functions are only applicable to a limited range of induction, the model cannot provide a solution for the case depicted in Figure 7f.

### 3.4 Comparison of results of the dynamic inflow model with CFD simulations

In this section, we compare the induction results obtained using the suggested dynamic inflow model (labeled *Ferreira*), the semi-free wake vortex ring model, and the actuator disc simulations in *OpenFOAM* (labeled *CFD*).

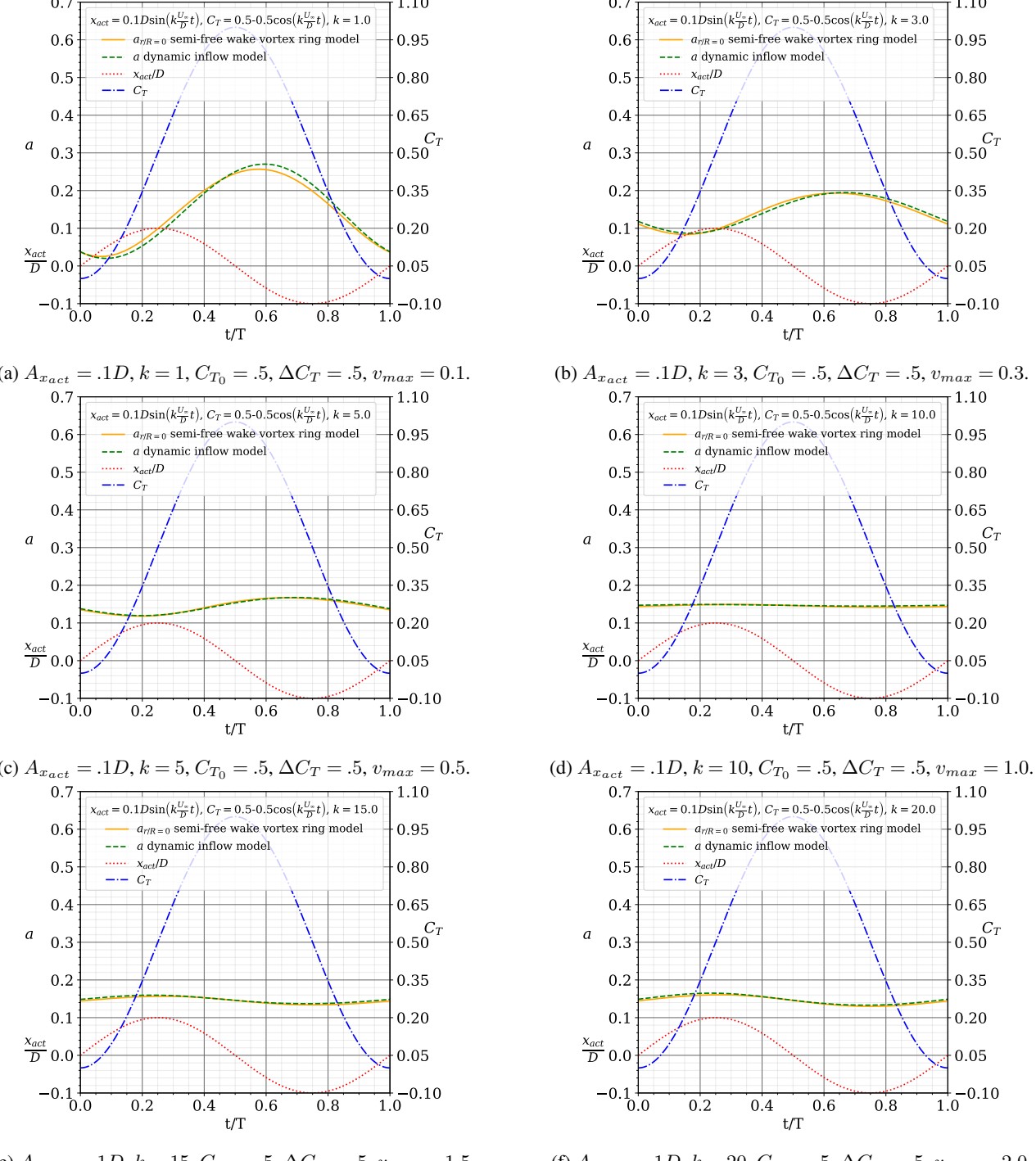

(a) $A_{x_{act}} = .1D$, $k = 1$, $C_{T_0} = .5$, $\Delta C_T = .5$, $v_{max} = 0.1$.

(b) $A_{x_{act}} = .1D$, $k = 3$, $C_{T_0} = .5$, $\Delta C_T = .5$, $v_{max} = 0.3$.

(c) $A_{x_{act}} = .1D$, $k = 5$, $C_{T_0} = .5$, $\Delta C_T = .5$, $v_{max} = 0.5$.

(d) $A_{x_{act}} = .1D$, $k = 10$, $C_{T_0} = .5$, $\Delta C_T = .5$, $v_{max} = 1.0$.

(e) $A_{x_{act}} = .1D$, $k = 15$, $C_{T_0} = .5$, $\Delta C_T = .5$, $v_{max} = 1.5$.

(f) $A_{x_{act}} = .1D$, $k = 20$, $C_{T_0} = .5$, $\Delta C_T = .5$, $v_{max} = 2.0$.

**Figure 4.** Comparison of the results of induction by the semi-free wake vortex ring model and the proposed dynamic inflow model at center of the actuator $r/R = 0$ for a sinusoidal surge motion with $x_{act} = A_{x_{act}} \sin\left(kU_\infty/Dt\right)$ (also plotted) and with $C_T = C_{T_0} - \Delta C_T \cos\left(kU_\infty/Dt\right)$ (also plotted). The results are plotted over one period, along the non-dimensioned time $t/T$. Cases with different reduced frequency.

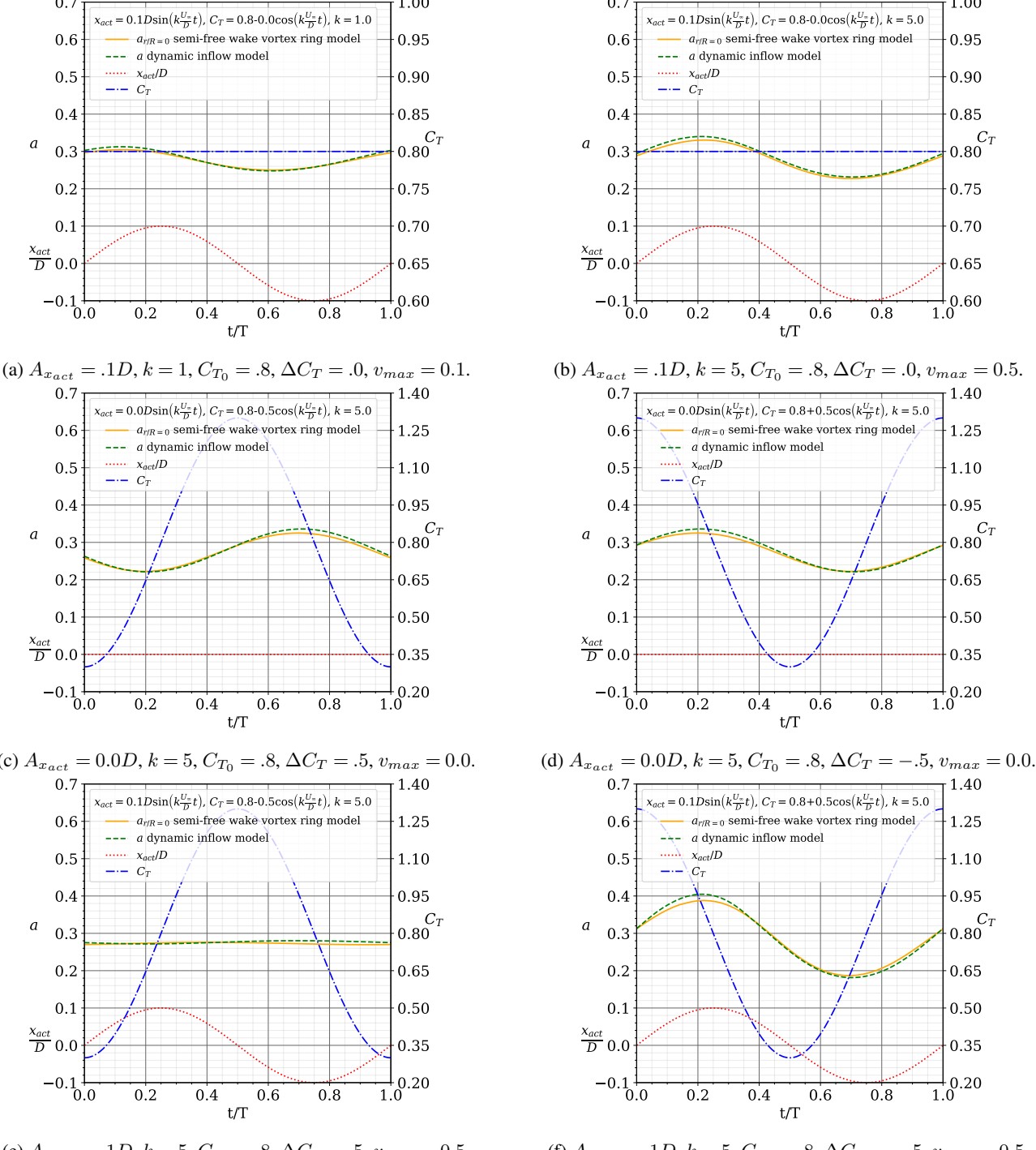

(a) $A_{x_{act}} = .1D$, $k = 1$, $C_{T_0} = .8$, $\Delta C_T = .0$, $v_{max} = 0.1$.

(b) $A_{x_{act}} = .1D$, $k = 5$, $C_{T_0} = .8$, $\Delta C_T = .0$, $v_{max} = 0.5$.

(c) $A_{x_{act}} = 0.0D$, $k = 5$, $C_{T_0} = .8$, $\Delta C_T = .5$, $v_{max} = 0.0$.

(d) $A_{x_{act}} = 0.0D$, $k = 5$, $C_{T_0} = .8$, $\Delta C_T = -.5$, $v_{max} = 0.0$.

(e) $A_{x_{act}} = .1D$, $k = 5$, $C_{T_0} = .8$, $\Delta C_T = .5$, $v_{max} = 0.5$.

(f) $A_{x_{act}} = .1D$, $k = 5$, $C_{T_0} = .8$, $\Delta C_T = -.5$, $v_{max} = 0.5$.

**Figure 5.** Comparison of the results of induction by the semi-free wake vortex ring model and the proposed dynamic inflow model at center of the actuator $r/R = 0$ for a sinusoidal surge motion with $x_{act} = A_{x_{act}} \sin{(kU_\infty/Dt)}$ (also plotted) and with $C_T = C_{T_0} - \Delta C_T \cos{(kU_\infty/Dt)}$ (also plotted). The results are plotted over one period, along the non-dimensioned time $t/T$. The results detail the separate effects of motion and load.

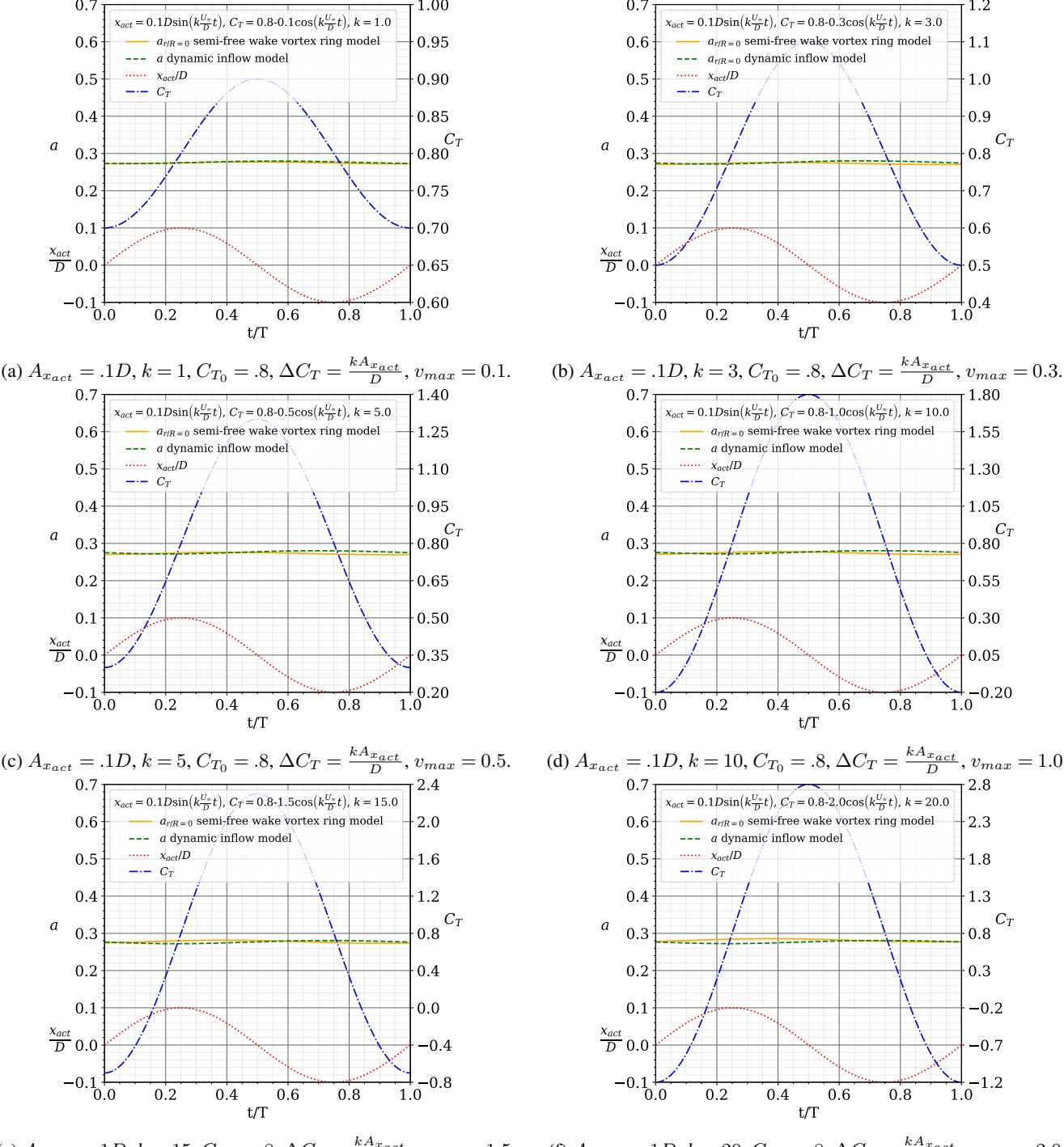

(a) $A_{x_{act}} = .1D$, $k = 1$, $C_{T_0} = .8$, $\Delta C_T = \frac{kA_{x_{act}}}{D}$, $v_{max} = 0.1$.

(b) $A_{x_{act}} = .1D$, $k = 3$, $C_{T_0} = .8$, $\Delta C_T = \frac{kA_{x_{act}}}{D}$, $v_{max} = 0.3$.

(c) $A_{x_{act}} = .1D$, $k = 5$, $C_{T_0} = .8$, $\Delta C_T = \frac{kA_{x_{act}}}{D}$, $v_{max} = 0.5$.

(d) $A_{x_{act}} = .1D$, $k = 10$, $C_{T_0} = .8$, $\Delta C_T = \frac{kA_{x_{act}}}{D}$, $v_{max} = 1.0$.

(e) $A_{x_{act}} = .1D$, $k = 15$, $C_{T_0} = .8$, $\Delta C_T = \frac{kA_{x_{act}}}{D}$, $v_{max} = 1.5$.

(f) $A_{x_{act}} = .1D$, $k = 20$, $C_{T_0} = .8$, $\Delta C_T = \frac{kA_{x_{act}}}{D}$, $v_{max} = 2.0$.

**Figure 6.** Comparison of the results of induction by the semi-free wake vortex ring model and the proposed dynamic inflow model at center of the actuator $r/R = 0$ for a sinusoidal surge motion with $x_{act} = A_{x_{act}} \sin(kU_\infty/Dt)$ (also plotted) and with $C_T = C_{T_0} - \Delta C_T \cos(kU_\infty/Dt)$ (also plotted). The results are plotted over one period, along the non-dimensioned time $t/T$. These cases are defined by $\Delta C_T = \frac{kA_{x_{act}}}{D}$.

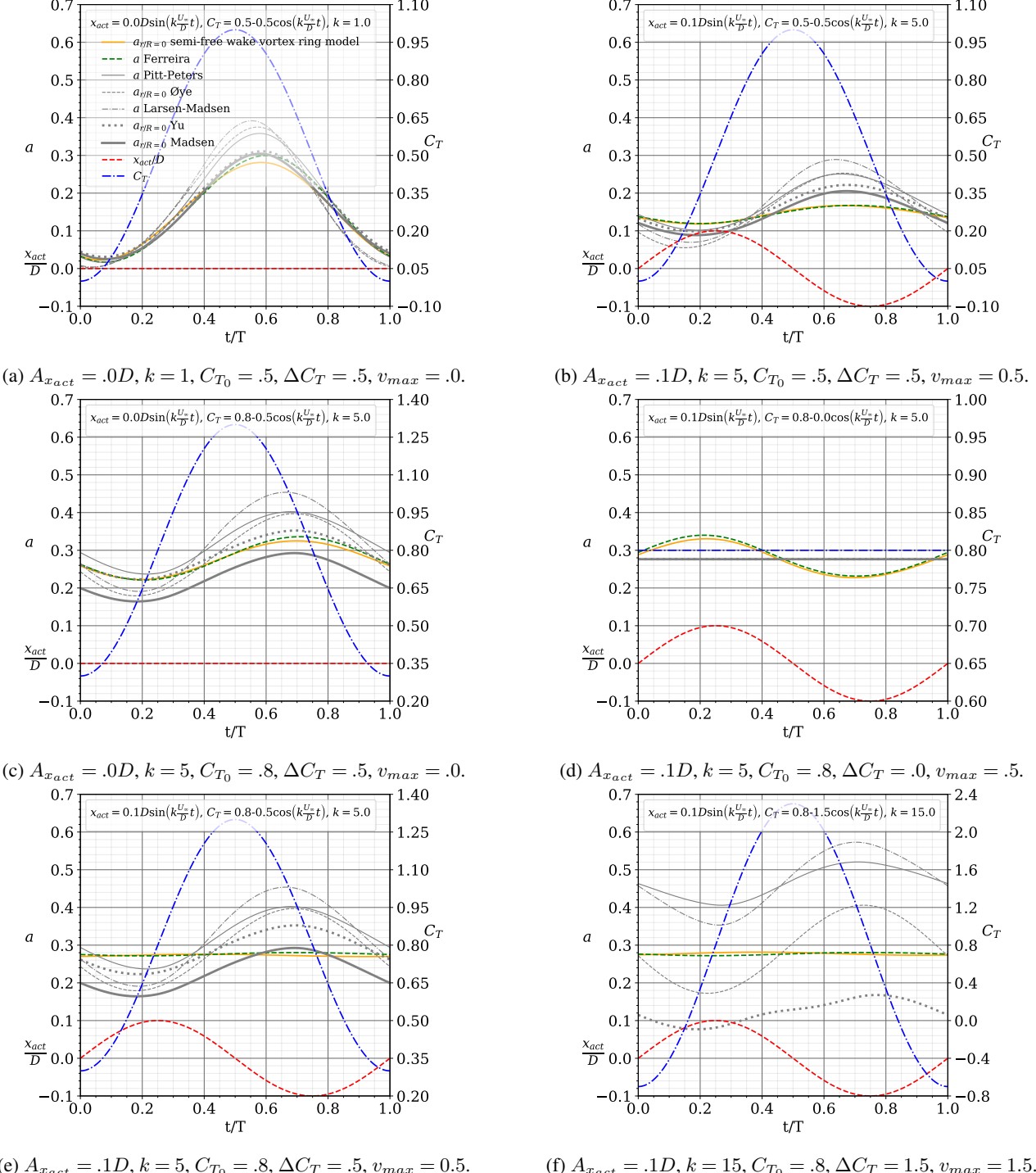

(a) $A_{x_{act}} = .0D$, $k = 1$, $C_{T_0} = .5$, $\Delta C_T = .5$, $v_{max} = .0$.

(b) $A_{x_{act}} = .1D$, $k = 5$, $C_{T_0} = .5$, $\Delta C_T = .5$, $v_{max} = 0.5$.

(c) $A_{x_{act}} = .0D$, $k = 5$, $C_{T_0} = .8$, $\Delta C_T = .5$, $v_{max} = .0$.

(d) $A_{x_{act}} = .1D$, $k = 5$, $C_{T_0} = .8$, $\Delta C_T = .0$, $v_{max} = .5$.

(e) $A_{x_{act}} = .1D$, $k = 5$, $C_{T_0} = .8$, $\Delta C_T = .5$, $v_{max} = 0.5$.

(f) $A_{x_{act}} = .1D$, $k = 15$, $C_{T_0} = .8$, $\Delta C_T = 1.5$, $v_{max} = 1.5$.

**Figure 7.** Comparison of the results of induction by the semi-free wake vortex ring model, the proposed dynamic inflow model *Ferreira*, and the *Pitt-Peters*, *Øye*, *Larsen-Madsen*, *Yu* and *Madsen* models for a sinusoidal surge motion with $x_{act} = A_{x_{act}} \sin\left(kU_\infty/Dt\right)$ (also plotted) and with $C_T = C_{T_0} - \Delta C_T \cos\left(kU_\infty/Dt\right)$ (also plotted). The results are plotted over one period, along the non-dimensioned time $t/T$.

Figure 8 compares the results of induction by CFD (black) and the semi-free wake vortex ring model (orange) at five radial positions $r/R = [0.0; 0.4; 0.6; 0.6; 0.9]$ (different line styles applied to the color black or orange, as defined in the legend), as well as the proposed 1D dynamic inflow model *Ferreira* (green) for various motion and thrust combinations.

The results indicate a high degree of agreement. For average $C_T = 0.5$ (Figures 8a and 8b), the CFD and semi-free wake vortex models produce induction differences of less than 0.01 at various radial places. Even for the case with motion (Figure 8b), the findings demonstrate a minor radial variation in induction. The dynamic inflow model agrees well with the higher-fidelity models. For typical $C_T = 0.8$ examples (Figures 8c to 8f), the CFD model and the semi-free wake vortex model agree very well in terms of the radial variation of the induction. Both models' findings demonstrate how the direction of induction's radial variation (increasing or decreasing radially) varies with the combination of loading and motion. Both models agree in the prediction of the phase and magnitude of this fluctuation. The absolute difference between the two models is their predicted time-averaged induction, with the semi-free wake model agreeing with the steady state solution and the CFD simulation being around $0.015 - 0.02$ less than the steady state solution. The one-dimensional dynamic inflow model agrees well with the higher-fidelity simulations. The near-wake effect justifies the radial variation. Its implementation in the model is deferred until more work is completed.

## 4   Conclusions

We devised, built, and validated a new dynamic inflow model capable of simulating the induction at an actuator disc during surge motion, thereby extending BEM's capability to simulate Floating Offshore Wind Turbines in large and fast surge motions. The new dynamic inflow model was tested against previous CFD simulations, new CFD simulations given in this work, and simulations using a semi-free wake vortex ring model. Additionally, these higher-fidelity models demonstrated the effect of motion and loading on induction's radial variation. To validate the model thoroughly, we examined situations with significant amplitudes of motion and load (e.g., twice the motion velocity and wind speed, and $DeltaCT = 2.0$), as well as phase-coupling between motion and load. In all scenarios tested, the results of the novel dynamic inflow model are in excellent agreement with those of the higher fidelity models. Additionally, the new dynamic inflow model was compared to several well-known and established dynamic inflow models.

The results demonstrated that the actuator's motion does not imply a turbulent wake or vortex ring state, even when the motion is significantly faster than the unperturbed wind speed. Previous pronouncements of this effect were based on an inaccurate interpretation of the actuator's accelerated reference frame.

Additionally, the results confirmed that, while increasing frequency of motion can result in increased loading and velocity amplitudes, the streamtube's inertia results in essentially constant induction. The effect of motion tends to cancel out the variation in thrust (assuming a $DeltaCT$ proportional to the surge velocity), and the variance in induction at the actuator decreases with greater frequency.

The model formulates wake generation and convection in lagrangian terms, and the resulting vorticity-velocity system solution of the induction field is frame-invariant. This allows the accelerating actuator's induction to be predicted. The model is based

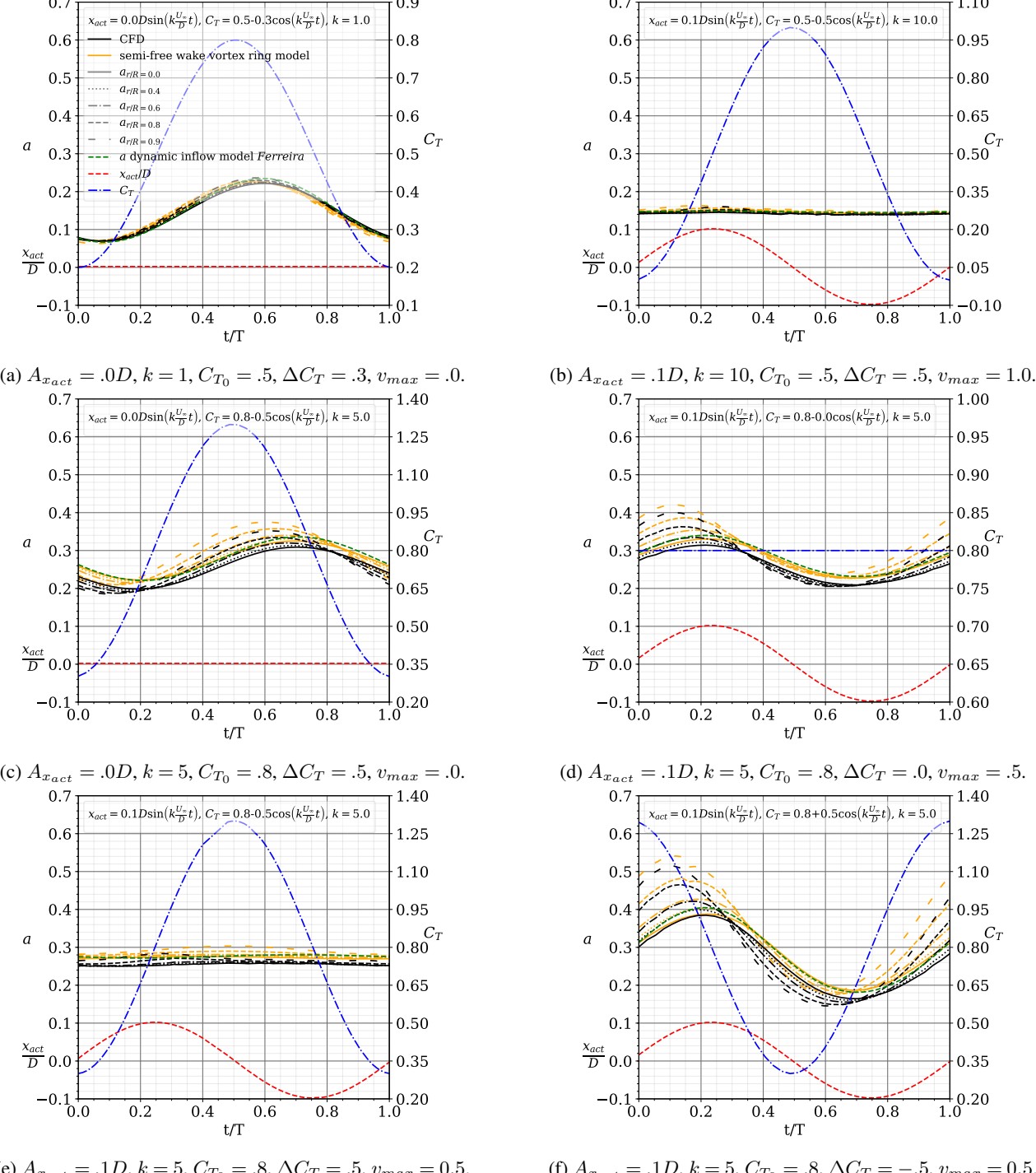

(a) $A_{x_{act}} = .0D$, $k = 1$, $C_{T_0} = .5$, $\Delta C_T = .3$, $v_{max} = .0$.

(b) $A_{x_{act}} = .1D$, $k = 10$, $C_{T_0} = .5$, $\Delta C_T = .5$, $v_{max} = 1.0$.

(c) $A_{x_{act}} = .0D$, $k = 5$, $C_{T_0} = .8$, $\Delta C_T = .5$, $v_{max} = .0$.

(d) $A_{x_{act}} = .1D$, $k = 5$, $C_{T_0} = .8$, $\Delta C_T = .0$, $v_{max} = .5$.

(e) $A_{x_{act}} = .1D$, $k = 5$, $C_{T_0} = .8$, $\Delta C_T = .5$, $v_{max} = 0.5$.

(f) $A_{x_{act}} = .1D$, $k = 5$, $C_{T_0} = .8$, $\Delta C_T = -.5$, $v_{max} = 0.5$.

**Figure 8.** Comparison of the results of induction by CFD and the semi-free wake vortex ring model at radial positions $r/R = [0.0; 0.4; 0.6; 0.6; 0.9]$ and the proposed dynamic inflow model *Ferreira* for a sinusoidal surge motion with $x_{act} = A_{x_{act}} \sin(k U_\infty / Dt)$ and with $C_T = C_{T_0} - \Delta C_T \cos(k U_\infty / Dt)$. The results are plotted over one period, along the non-dimensioned time $t/T$.

on the well-established techniques developed by Øye (1986), Larsen and Madsen (2013), Madsen et al. (2020), and Yu (2018)
and Yu (2018).

The straightforward approach is simply implementable in BEM models. The existing implementation already addresses the scenario of heavily loaded streamtubes; yet, even for static actuators, this region remains challenging. For future work, the model's simplicity and analytical formulation make it well-suited for optimizing and controlling FOWTs. The prediction of induction at the tip region is postponed till further research is completed.

**Appendix: Appendix A: Implementation of the proposed dynamic inflow model in Python.**

```python
import numpy as np

def dynamic_inflow_model_Ferreira_moving_actuator(CT, Uinf, Uref, R, dt, u_act, u_str,
                                                  v_actuator=0, glauert=False, dynamic=False):
    # calculates the induction velocity at the center of
    # an actuator disc in surge motion
    # developed by:      Carlos Ferreira,
    #                                Delft University of Technology, October 7th 2020
    # inputs:
    #           CT - thrust coefficient of the actuator
    #           Uinf - unperturbed wind speed
    #           Uref - reference unperturbed wind speed in the inertial reference
    #                   that contains the streamtube
    #           R -  radius of the actuator
    #           dt - delta time
    #           u_str - streamtube induction velocity (changed in function)
    #           u_act - induction velocity at actuator (changed in function)
    #           v_actuator - velocity of the actuator, default value=0
    # outputs:
    #           u_str - streamtube induction velocity (changed in function)
    #           u_act - induction velocity at actuator (changed in function)
```

```python
        # define length scales for actuator/near wake scale
        # and streamtube/far wake scale
        len_act = 1.*R
        len_str = 5.*R

        if dynamic:
            # update the reference unperturbed wind speed taking into account
            # the motion of the actuator
            expf=np.exp(-dt*Uref/len_str)
            Uref = Uref*expf+(Uinf-v_actuator)*(1-expf)
        else:
            # define the reference velocity as the same as the unperturbed wind speed
            Uref = Uinf

        #calculate reference streamtube velocity
        Ustr=Uref-(u_act+u_str)/2

        # calculate value of forcing function for a
        Uqs=np.array(CT/4*Uinf**2/Ustr)

        # apply adapted Glauert correction if required
        Induction_Glauert=   1-np.sqrt(1.816)/2;

        if glauert:
            IndGlauert= np.logical_and(Ustr<(Uref*(1-Induction_Glauert)),Uqs>0)
            # IndGlauert -> index of cases that the reference streamtube velocity
            # is lower than the criteria by Glauert and Uqs positive
            Uqs[IndGlauert]=-1.88254912-1.54029217*Uqs[IndGlauert]**(1/2) +\
            4.08622347*Uqs[IndGlauert]**(1/4) # from curve fit of Glauert's
                                              # correction for heavy loaded flow

        # define time scales of convection of the wake for actuator/near wake scale
```

```python
        # and streamtube/far wake scale. We define them as the inverse of the time
        # scale, to avoid divide by zero due to the velocity of the actuator

        # time of relative convection of old near actuator solution
        inv_tau_act_1 = (Uinf-0.5*u_act-v_actuator)/len_act
        # time of convection of the new generated wake in relation to the actuator
        inv_tau_act_2 = (Uref-0.5*u_act)/len_act
        # time of convection of the old and new wake at stramtube/far wake scale
        inv_tau_str = (Uref-0.5*u_str)/len_str

        # calculate new values of the induction velocity at actuator
        # and streamtube induction velocity
        u_act = u_act*np.exp(-dt*inv_tau_act_1)+Uqs*(1-np.exp(-dt*inv_tau_act_2))
        u_str = u_str*np.exp(-dt*inv_tau_str)+Uqs*(1-np.exp(-dt*inv_tau_str))

        return u_act, u_str, Uref
```

*Author contributions.* CF imagined, proposed, and developed the ideas, reviewed previous work, proposed the hypothesis and methods, derived and programmed all models, created the simulations, the results and conclusions, and wrote the majority of the text. WY is an expert in dynamic inflow models who provided models for verification and checked consistency of the concept, discussed the idea, and reviewed previous work. AS developed the CFD simulations in OpenFOAM and provided key elements of the literature review, in particular the review of case studies by other authors. AV is a leader on FOWTs, discussed the ideas, and reviewed previous work. All authors contributed to the review process, both scientific and editorial.

*Competing interests.* No competing interests are present.

*Acknowledgements.* CF would like to acknowledge Jing Dong from Delft University of Technology for the stimulating debate who sparked his interest in this topic. Her research was a key trigger for the development of this work.

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
