# Peer review of "Dynamic inflow model for a Floating Horizontal Axis Wind Turbine in surge motion"

_Wind Energy Science, 2021_

## Referee Comment (RC2)

[referee-annotated manuscript omitted]

---

## Author Comment (AC1)

Referee review of "Dynamic inflow model for a Floating Horizontal Axis Wind Turbine in surge motion" by Ferreira et al.

**Referee's comment** *The paper deals with a timely and interesting set of questions, related to the state of actuator disk/mo-*
600   *mentum theory for the case of oscilatory disk motions. Clearly this area is of interest for wind turbines placed on off-shore*
*platforms that will oscillate back and forth and will change the inflow velocity being seen by the system. The overall conclu-*
*sions, which this referee finds reasonable and interesting, is that if properly formulated, standard actuator disk approach still*
*works, as long as the correctly chosen U-infinity(t) is used. The introduction and motivation are well described and the survey*
*of prior work (in particular Fig 1) is very good. The introduction also gives the impression that a more fundamentals oriented*
605   *rational method will be proposed to deal with non-inertial to inertial reference frames etc etc. that has caused confusions in*
*the past. So that all seemed very promising.*

       **Answer:** Thank you for this positive starting comment. The objective of the work is to derive a model based on correct physics.

**Referee's comment** *However, once the "meat" of the contribution starts being described, the material is suddenly presented*
610   *as an "algorithm" to be implemented in python, etc. and there seems to be no connection whatsoever with any actual physics*
*or principles being invoked. That is to say, where did Eqs. 20 and associated Eqs. for $u_act, u_str$ etc, Eqs. 17 & 18 come from?*
*There seems to be no connection with any actual physics or principles being invoked.*

       **Answer:** Thank you for this helpful comment. The expression "algorithm" has been replaced by model. More impor-
tantly, text has been added/modified to explain the derivation of the equations.

615   **Referee's comment** *More specifically, in line 195 authors claim to be computing "new solutons for the streamwise induction*
*velocity at actuator". What equation is being solved exactly and how is the solution obtained? Up to this point in the paper*
*there is not a single dynamical evolution equation being presented. One would expect some equation of the form $du/dt = ...$*
*and then the solution is Eqs 17,18 etc. Instead, what the authors seem to be doing is simply a-priori assuming that a time*
*filtering will have benefits of some sort to be used as inflow for the model implementation to come later, but it does not look*
620   *like Eqs. 17 and 18 are "solutions" to anything in particular. Only in point 6 of the introductory sentences there is a reference*
*to a time-filtering method (Larsen-Madsen model). In that paper the time-filtering was motivated simply by saying something*
*along the lines of "engineering model for response functions" including inertia of structures etc. How is that approach really*
*justified in light of the very fundemantal sounding comments made in the introduction of the paper? This paper should provide*
*a clear discussion of these aspects.*

625          **Answer:** Several points are mentioned in this comment. We will aim to address all. The model aims to present an
equivalent solution of the vorticity-velocity problem, in the perspective vorticity is shed at each time step and previously shed
vorticity is convected away from the actuator at each time step. This can be approximated by a convolution of the current
solution and a new steady state solution. This approach is on the basis of dynamic inflow models such as the one by Øye
(1986), Larsen and Madsen (2013), Yu (2018) and Madsen et al. (2020). These models often referred to a filtering approach
630   of the near and far wake, which is a reasonable description; we opted for the same description, but the language is not totally

correct. The text has been modified to avoid the word "filter" and instead present the evolution from one vorticity system to a new vorticity system.

Regarding the point of the need of an explicit $du/dt = ...$ formulation, here one politely disagrees with the referee. The dynamic inflow model by Pitt and Peters (1981) (and also the ECN model) has an explicit time integration of du/dt because it

635 models a linearized form of the unsteady momentum equation. The model of Øye (1986), although it presents a $du/dt = ...$ formulation, is in fact solving the same convolution problem as the models by Larsen and Madsen (2013), Yu (2018) and Madsen et al. (2020), just with a different numerical integration procedure. The formulation of solutions that decay with time through an exponential of time (as these last cited models and the model proposed in this work) provides an implicit form of time integration and, a clearer interpretation of the phenomena. However, the equation presented in this work can be converted

640 to a $du/dt = ...$ formulation as Øye (1986). We have added/modified the text to make this clearer.

**Referee's comment** *Presentation of results (Figs. 3-5) show one cycle of resulting induction factor for various conditions and good results compared with the semi-free wake vortex ring model are shown. Was the inflow velocity time-filtering approach simply proposed by noting empirically from such plots that time-filtering the input would yield desired results? And parameters obtained by fitting the observed behaviors? That may be a fine approach for very applied settings, but unless better justified*

645 *by analysis of governing equations, it it does not seem to rise to the level of a scientific contribution since it does not seem convincing that it can be generalized in any way to other conditions.*

**Answer:** Choice of the formulation of the model was not based on what works. Once it was defined that the model needs to evaluate the solution in the inertial reference frame and accounts for the motion of the actuator, it was necessary to have a formulation that was invariant with the reference frame, and that is the vorticity-velocity formulation. The model needs

650 to account for the change of the vorticity system, as new wake is shed and old wake is convected, and the relative position of the actuator in relation of the vorticity system. The text was modified to better explain this.

**Referee's comment** *In view of the above comments, it is recommended that the authors aim to justify and derive the "time-filtering" approach somehow, if that is possible. If not possible, publication in WES is perhaps not fully justified and also, then the characterization of prior work (references to past "confusions") should be reworded to avoid raising the readers' hopes*

655 *that the present paper will clarify these things.*

**Answer:** The changes to the text should clarify the physics behind the derivation of the model.

*Some additional comments for minor revisions, if useful:*

**Referee's comment** *Abstract, first sentence: the statement " ..surge motions .. when faster than the local wind speed, cause rotor-wake interaction." Do the authors mean to imply that only if surge motion is larger than, say, 8 m/s (local wind speed),*

660 *there will be rotor-wake interactions? One would expect "interactions" even at much lower surge motion speeds.. Needs more precise wording. It seems when authors say "interactions" they have something very specific in mind but at this stage of the paper readers will have more general interpretations of "interactions" in mind.*

**Answer:** The text is modified to specify *blade-vortex interaction*. The abstract is also revised.

**Referee's comment** *Line 24: do the authors mean to say "a turbulent wake with the wake in front of the turbine?" since the*

665 *normal state of turbine wakes is a turbulent wake state in the first place.*

**Answer:** The precise sentence written above was not part of the text. For clarification, "turbulent wake state" describes the streamtube loading condition and wake-breakdown/flow reversal downwidn of the rotor. That is not the normal state of a wind turbine wake (which is turbulent, but not in "turbulent wake state").

**Referee's comment** *Sentences are often unclear referring to undefined properties that are perhaps coming later? Text needs*
670 *careful proof-reading for such things. For instance, line 171, there is talk about "to be used later as a forcing function for the filter functions". At this stage of the paper, it is unclear what filtering functions this refers to. Again, wordings need to be critically reviewed throughout.*

**Answer:** The text, and in particular the section mentioned, has been changed and reviewed.

**Referee's comment** *I found the set of 9 "hypothesis" (lines 85-110) a bit tedious to go through, some read like the con-*
675 *clusions of which one is not yet convinced without reading the rest of the paper, others read like additional assumptions, etc. They really read like sentences in a research proposal and seem suboptimal at this place in the paper. I would recommend restructuring/shorten/or even delete 85-110.*

**Answer:** The text was modified.

---

## Author Comment (AC2)

**Referee's comment** *In this paper the authors present a dynamic inflow model suitable for FOWT, and verify the results against high and mid fidelity simulations. This is a nicely written paper, with interesting methods and conclusions. I have some general comments that I hope can improve the revision of the paper.*

        **Answer:** Thank you for the kind comment. You comments have been very useful towards improving the work. Thank you.

**Referee's comment**

     - I believe the paper would benefit from adding more justifications for each of the important equations of the model. You'll find several specific comments in the pdf regarding this.*My general comments are the following:*

     *- I believe the paper would benefit from adding more justifications for each of the important equations of the model. You'll find several specific comments in the pdf regarding this.*

*Answer: The text ws modified to address this, including the comments in the pdf, which are listed below.*

     *Referee's comment - Some results for various radial positions would probably be needed to support the conclusion that the model compare well with the ring model for up to r/R=0.8.*

        *Answer: Section 3.4 was added, where the model is compared with CFD simualtions and semi-free wake vortex model simulations, includign results at different radial positions. These results are used to support the discussion and conclusion,*

*which are modified.*

     *Referee's comment*

     *- Comparison with similar models: How does the model compare with the model of Oye, and Hawc2? All models use two time constants. Oye's model has the advantage of being continous. - I would suggest adding a discussion section to address the following points:*

*- Comparison with similar models: How does the model compare with the model of Oye, and Hawc2? All models use two time constants. Oye's model has the advantage of being continous.*

        *Answer: Section 3.3 was added, where the proposed dynamic inflow model is compared with several other dynamic inflow models, namely the one by Pitt and Peters (1981) as described by Yu (2018), by Øye (1986) as described by Yu (2018), the model by Larsen and Madsen (2013), the model by Yu (2018) (also described by Yu et al. (2019b)) and the model presented*

*in the work of Madsen et al. (2020).*

     *Referee's comment - What are the limitation of the current model towards the tip? How could these be lifted?*

        *Answer: Once again, we refer to the new Section 3.4 .*

     *Referee's comment - Vortex ring state: The paper mention that vortex ring states do not occur as commonly thought, but I think this might need further justifications. The paper demonstrates that at high frequencies, the variation of inductions are*

*limited, but variations are expected for lower frequencies. The cases studied in this paper were reasonably far from "high thrust" conditions. I think it would be worth investigating the variation of amplitudes of "a", for various "k" and "CT", and try to reach the vortex ring state. There has to be a point where the vortex ring state will be reached. (Obviously, this will likely go*

*beyond the region of validity of the model and the vortex-ring-based models, so it will have to be treated with care – I do not expect the vortex-ring based model to accurately capture the vortex-ring state which will be highly turbulent and diffusive.).*

*The question that could be answered and would be really interesting would be whether the vortex ring state model occurs "sooner" (for some low frequencies maybe) than one would expect from the steady conditions (zero frequency), or "later", or simply "at the same time". I think such an investigation will really add to the paper (again, keeping the limitations of both models in mind). At least a small moderation on the fact that the vortex ring state was not really "tested" would be great (I understand that the study still makes a point that it was not reached for "moderately loaded" rotors).*

***Answer:*** *Previous authors claimed that high thrust coefficients occurred because the perceived velocity in the reference frame of turbine becomes very low or negative, and that this represented a vortex ring state . That interpretation is incorrect. However, regardless of the motion, the streamtube can enter vortex ring state if a large loading is applied for a long enough time. So, the work does not mean that vortex ring state cannot occur, only that the interpretation of the velocity perceived in the reference frame of the wind turbine does not represent vortex ring state. The text is modified to further clarify* 725 *this.*

***Referee's comment***

*Congratulation for your work, I'll be looking forward to review a revised version of this paper.*

*Emmanuel I enclose some specific comments (along the lines of my general comments) in the pdf enclosed.*

*Congratulation for your work, I'll be looking forward to review a revised version of this paper.*

*Emmanuel*

***Answer:*** *Thank you very much for the additional annotations and the overall appreciation. The answers to the comments in the pdf can be found below.*

***Annotations by second reviewer***

***Referee's comment*** *suggest stressing again here that vact is constant (time invariant). (note on p.2)*

***Answer:*** *Thank you for the very good suggestion. The text has been added explaining Equation 2 is only valid when* $v_{act}$ *is constant.*

***Referee's comment*** *suggest: arbitrary or periodic (note on p.3)*

***Answer:*** *Thank you for the very good suggestion. The text was modified.*

***Referee's comment*** *I would suggest using small omega to avoid confusion with Omega typically used for rotor speed. The* 740 *context is yet clear in this paper. (note on p.5)*

***Answer:*** $\Omega$ *was replaced to* $\omega$

***Referee's comment*** *How realistic is it to assume a uniform and sinusoidal CT distribution? I'm guessing you have found this to be true using higher fidelity/vortex method. Could you discuss/mention this a bit here? (note on p.5)*

***Answer:*** *Thank you for this observation. It also connects with the next observation. The following text was added: The* 745 *sinusoidal loading approximates the load oscillations observed by other authors, as described in Section... . The load change is a first-order result of the sinusoidal change in the non-entry boundary condition on the blades/actuator surface caused by the sinusoidal motion (this is further expanded in Section... ).*

*Referee's comment* You can maybe add here the formula that supports this sentence (I'm a formula person..) (note on p.5)

Answer: Thank you for the suggestion. Equation7 was added and the text was extended to explain it.

*Referee's comment* It took me a bit of time to understand this figure. Could it maybe be made clearer in the text that this figure simply shows what are the "operating conditions" tested in the literature. (note on p.6)

Answer: The caption of the figure was changed to indicate this.

*Referee's comment* I believe the model of Oye (found also in the book of Martin Hansen) also uses two time scales, and predates these references. (note on p.7)

Answer: The reviewer is absolutely correct. A reference to the earlier work by Øye has been added. The Øye model is also used in Section 3.3.

*Referee's comment* Is this model not also inspired by the one from de Vaal? (note on p.7)

Answer: The simulations by de Vaal were in Fluent. Or is the reviewer suggesting another reference? The text is not changed.

*Referee's comment* Potentially use vact in this formula (note on p.8)

Answer: The formula is correct according to the derivation. It is not a typo. The formula was not changed.

*Referee's comment* Can you mention how this formula was obtained as a quasi-steady solution? (note on p.8)

Answer: The text has been modified to explain the formula more clearly. The formula is an adaptation of the 1D actuator disc thrust equation, where the term of mass flow rate is changed to the weighted term.

*Referee's comment*  Could you justify the use of this formula? For an actuator disk moving against the wind, I would think the convection velocity would be Uinf - uact - ustr/2, no? Maybe this could be mentioned/discussed in the text. (note on p.8) Coming back up here, I noticed that you have both the notion of uact and vact. It was not clear to me that there was a distinction between the two. What is meant by the induction velocity of the actuator disk? (Similarly, the other terms in this equation might need to be clearly introduced and defined to avoid confusion). (note on p.8)

Answer: The text has been expanded to include a more detailed explanation. $u_{act}$ is the induction at the location of the actuator in the reference frame of the reference wind speed. the velocity of motion of the actuator is defined as $v_{act}$.

*Referee's comment*  It was not clear to me that this was not already the case. Could you stress above (or using subsections) that the first developments are for a constant vact? (note on p.9)

Answer: The derivation of the model was for the case of a oscillatory motion (average displacement is zero). The additional equation allows to consider a reference frame of unperturbed wind speed and an actuator motion which as a non-zero average displacement (e.g. forward motion plus oscillatory motion). The text was modified to make this clearer.

*Referee's comment*

I believe Oye uses $u_int$ for instance. (note on p.9) It seems that uact is actually an intermediate induced velocity. Can you give a physical meaning to this velocity? I would suggest another notation, because uact has been confusing me above, it can easily be confused wih vact.

I believe Oye uses $u_int$ for instance. (note on p.9)

**34**

*Answer:* $u_{act}$ *is not an intermediate velocity, it is the induction at the actuator. the definition of* $u_{act}$ *was edited to be made clearer.*

*Referee's comment Could these equations be written in continuous form? (like Oye) (note on p.9)*

*Answer: Yes. But this formulation has an higher order of numerical integration.*

*Referee's comment It might be worth (somewhere in the text) to mention how this formulation differs from Oye's formulation, and Hawc2 formulation. My first impression is that they are very similar, modulo some scaling and definitions of time constants. (note on p.9)*

*Answer: The comment is correct. A text referencing this was added.*

*Referee's comment More justifications would be needed here the choices do not appear straightforward to me. Could you discuss/justify them? Could you mention why were the induced velocity are not used in the time constants for instance? (note on p.9)*

*Answer: Text was added to justify the lengths scales as relations to the scales of wake expansion and vorticity-velocity solution system. The second question of the reviewer is not clear, as the induction velocity is used to determine the time scales.*

*Referee's comment Potentially mention in parenthesis the sign of vact. (note on p.9)*    *Answer: Added to the text.*

*Referee's comment Could you precise in the text which reference velocity is used to define a and CT? (note on p.10)*

*Answer: All values are defined in relation to* $U_\infty$. *Text was added to this effect.*

*Referee's comment Could the results of the dynamic inflow model be plotted at different radial position too? (note on p.10)*

*Answer: The formulation of the dynamic inflow model is 1D. The radial variation, which is modelled in other dynamic*

*inflow models, has not been translated to this new model. That topic is left for future research.*

*Referee's comment Is there a reason for this choice? How does the model perform at other radial stations? Stronger induction effects might be found at larger radial position (closer to the wake). Could you show a small study for different radial position? (note on p.11)*    *Answer: As in the previous comment, the dynamic inflow model is 1D. In the text, the reference to "center of the actuator" was removed.*

*Referee's comment It might be worth stressing in the figure which velocity is used to define a and CT. (note on p.12)*

*Answer: Text was added to the effect.*

*Referee's comment It might be worth discussing what's "wrong" with the model towards the tip. (note on p.17)*    *Answer: This was addressed in a previous comment. The text has been modified to address this.*

*Referee's comment I don't think this was presented in the paper, or I might have missed it. Presenting some results for this*

*would be great. (note on p.17)*    *Answer: This was presented in Section 3.1. However, the text was modified for clarity.*